# Microvascularization and Expression of Fibroblast Growth Factor and Vascular Endothelial Growth Factor and Their Receptors in the Mare Oviduct

**DOI:** 10.3390/ani11041099

**Published:** 2021-04-12

**Authors:** Pedro Pinto-Bravo, Maria Rosa Rebordão, Ana Amaral, Carina Fernandes, António Galvão, Elisabete Silva, Pedro Pessa-Santos, Graça Alexandre-Pires, Rosário P. Roberto da Costa, Dariusz J. Skarzynski, Graça Ferreira-Dias

**Affiliations:** 1CERNAS (Research Center for Natural Resources, Environment and Society), Polytechnic Institute of Coimbra, 3045-601 Coimbra, Portugal; pbravo@esac.pt (P.P.-B.); ross@esac.pt (R.P.R.d.C.); 2Coimbra College of Agriculture, Polytechnic Institute of Coimbra, 3045-601 Coimbra, Portugal; milorebordao@gmail.com; 3CIISA—Centre for Interdisciplinary Research in Animal Health, Faculty of Veterinary Medicine, University of Lisbon, 1300-477 Lisbon, Portugal; nita.amaral@gmail.com (A.A.); fachica@hotmail.com (C.F.); elisabetesilva@fmv.ulisboa.pt (E.S.); gpires@fmv.ulisboa.pt (G.A.-P.); 4Institute of Animal Reproduction and Food Research, Polish Academy of Science, 10-748 Olsztyn, Poland; a.galvao@pan.olsztyn.pl (A.G.); d.skarzynski@pan.olsztyn.pl (D.J.S.); 5Hospitals of the University of Coimbra, 3004-561 Coimbra, Portugal; p.pessa59@gmail.com

**Keywords:** oviduct, mare, vascularization, vascular endothelial growth factor, fibroblast growth factor, fibroblast growth factor receptor, vascular endothelial growth factor receptor, kinase insert domain receptor, angiogenic factors

## Abstract

**Simple Summary:**

The oviduct provides the ideal conditions for fertilization and early embryonic development. Adequate vascularization is essential for proper oviduct physiological function. In this work on the mare oviduct, differences in the oviductal artery and arterioles and their ramifications in the infundibulum, ampulla and isthmus were examined. Locally, vascularization is modulated by the action of angiogenic factors, mediated by their specific receptors. In the present study, the isthmus presented the largest vascular area and the highest number of vascular structures in the follicular phase. We have also shown that the relative abundance of angiogenic transcripts and proteins, such as fibroblast growth factor 1 (*FGF1)* and 2 (*FGF2)* and vascular endothelial growth factor (*VEGF)*, and their respective receptors (*FGFR1*, *FGFR2*, *VEGFR2 = KDR*), were present in all portions of the oviduct throughout the estrous cycle. There was an increase in the transcripts of angiogenic receptors *FGF1* and *FGFR1* in the ampulla and isthmus, and of *FGF2* and *KDR* in the isthmus. This was also observed in the isthmus, where the relative abundance of proteins FGFR1 and KDR was the highest. This study shows that the equine oviduct presents differences in microvascular density in its portions. The angiogenic factors VEGF, FGF1, FGF2 and their respective receptors are expressed in all studied regions of the mare oviduct, in agreement with microvascular patterns.

**Abstract:**

The oviduct presents the ideal conditions for fertilization and early embryonic development. In this study, (i) vascularization pattern; (ii) microvascular density; (iii) transcripts of angiogenic factors (*FGF1*, *FGF2*, *VEGF*) and their receptors—*FGFR1*, *FGFR2*, *KDR*, respectively, and (iv) the relative protein abundance of those receptors were assessed in cyclic mares’ oviducts. The oviductal artery, arterioles and their ramifications, viewed by means of vascular injection-corrosion, differed in the infundibulum, ampulla and isthmus. The isthmus, immunostained with CD31, presented the largest vascular area and the highest number of vascular structures in the follicular phase. Transcripts (qPCR) and relative protein abundance (Western blot) of angiogenic factors fibroblast growth factor 1 (*FGF1)* and 2 (*FGF2)* and vascular endothelial growth factor (*VEGF)*, and their respective receptors (*FGFR1*, *FGFR2*, *VEGFR2 = KDR*), were present in all oviduct portions throughout the estrous cycle. Upregulation of the transcripts of angiogenic receptors *FGF1* and *FGFR1* in the ampulla and isthmus and of *FGF2* and *KDR* in the isthmus were noted. Furthermore, in the isthmus, the relative protein abundance of FGFR1 and KDR was the highest. This study shows that the equine oviduct presents differences in microvascular density in its three portions. The angiogenic factors VEGF, FGF1, FGF2 and their respective receptors are expressed in all studied regions of the mare oviduct, in agreement with microvascular patterns.

## 1. Introduction

In the mammalian female, the oviduct bridges the ovary to the uterus. In addition to the physical transport of the gametes and embryo, the oviduct provides paracrine signaling and biophysical support for sperm selection, capacitation and hyperactivation, fertilization, early preimplantation development and remodeling the epigenetic landscape of the embryo [1,2,3,4,5]. The ovarian and uterine arteries are responsible for blood supply to the oviduct, but substantial variations exist in the relative contributions of both arteries among species, individuals and hormonal influences [6]. As reported, the circulatory blood transports some of the components that cross epithelial oviductal cells to form the oviductal fluid [5]. In addition to providing adequate vascularization to the female reproductive tract, angiogenesis is crucial for the regulation of physiological processes related to reproductive function [7,8]. Angiogenesis consists on the growth of blood vessels from the existing vasculature, which occurs throughout the lifetime in both healthy conditions and in disease [7,9]. This is a very complex process. In the reproductive system, including the oviduct, angiogenesis is modulated by angiogenic/growth factors [7,8,10], anti-angiogenic factors [11,12,13], reproductive hormones [14,15], eicosanoids [9,16], nitric oxide [16,17], tumor necrosis factor α (TNFα) [18], oxygen (O_2_) content [19,20], and other factors [21,22,23]. Among a plethora of angiogenic/growth factors are vascular endothelial growth factor (VEGF) and its receptors 1 (VEGFR1) and 2 (VEGFR2, also known as kinase insert domain receptor—KDR) and fibroblast growth factor 1 and 2 (FGF1 and FGF2) and their receptors FGFR1 and FGFR2 [24,25,26,27,28]. It is worth noting that in the fallopian tubes of women, during early pregnancy recognition, anti-angiogenic genes are upregulated [29].

The angiogenic factor VEGF covalently binds to its receptor (VEGFR) through the extracellular receptor domain, promoting the recruitment of a variety of signaling molecules to VEGFR dimers, forming large molecular complexes that activate distinct cellular pathways [30]. The level of mRNA and the protein abundance of VEGF increase in the human oviduct in the peri-ovulatory period, mainly in the infundibulum and ampulla [14]. This upregulation of the oviductal VEGF system in women and in cows appears to be stimulated by gonadotrophins [14,31]. In the sow oviduct, epidermal growth factor (EGF), VEGF, FGF and their receptors are estrous-cycle- and implantation-dependent [15]. The VEGF system seems to be involved in a putative paracrine network that is necessary for proper fertilization and gamete transportation to successfully establish and maintain pregnancy in pigs [15,32].

Different members of the FGF family, mostly FGF1 and FGF2, under in vitro conditions can induce a complex pro-angiogenic phenotype, including proliferation, migration, protease production, integrin and cadherin receptor expression and intercellular gap-junction communication [33]. Thus, FGF1 and FGF2 specifically bind to cell surface-expressed receptors equipped with receptor tyrosine kinases, denoted as FGFR1 and FGFR2. Then, the activation of receptor kinase activity allows for the coupling to downstream signal transduction pathways that regulate proliferation, migration and differentiation of endothelial cells. FGFRs are involved with the activation of several parallel signaling pathways [34]. The presence of FGF family members in ovarian follicles of cows may be involved in the development of the pre-ovulatory follicle by stimulation of angiogenesis and granulosa cell survival and proliferation [35].

Interestingly, intimate crosstalk may exist among FGFs and several members of the VEGF family during angiogenesis. Several reports indicate that FGF2 itself can induce neovascularization indirectly by activating the VEGF/VEGFR system [36]. Since we have shown the importance of angiogenesis in endometrial and luteal function in mares [16,17,26,27,28,37,38], we have hypothesized that angiogenesis may also play a role in the mare oviduct. Therefore, the objectives of this study were to assess in the three portions (infundibulum, ampulla and isthmus) of the mare oviduct (i) the vascularization pattern; (ii) the microvascular density; (iii) the gene expression of angiogenic factors (*FGF1*, *FGF2* and *VEGF*) and their receptors—*FGFR1*, *FGFR2* and *KDR* respectively—and (iv) the protein expression of FGFR1, FGFR2 and KDR.

## 2. Materials and Methods 

### 2.1. Collection of Mare Oviducts

From early April to late September, from 40 randomly designated cyclic mares (3 to 9 years old), blood samples and internal genitalia were obtained *post-mortem* at the abattoir. Mares were handled and euthanized according to the Portuguese mandate (DL 98/96, Art. 1º) and European Legislation concerning animal welfare in stunning and euthanasia (EFSA, AHAW/04-027). 

Based on ante-mortem and post-mortem veterinarian examinations, only oviducts retrieved from healthy mares assigned for human consumption were used for the assays. Since the exact reproductive status of the mares was unknown, the various stages of the estrous cycle (follicular, early and mid-luteal stages) were identified based on the characteristics of the ovarian structures (follicles and corpora lutea) and confirmed by plasma progesterone (P_4_) concentrations, as described in [16,17]. Based on uterine swabs collected for the isolation of bacteria and cytologic evaluation, oviducts from mares with apparent reproductive problems, such as endometritis, were discarded from the study, as described in [39]. The oviducts from the ipsilateral side to the predominant ovarian structure in the follicular phase (FP), early-luteal phase (ELP) or mid-luteal phase (MLP) [16,17] were collected for the different assays. Samples from oviduct segments (infundibulum, ampulla and isthmus) were placed in (i) RNAlater (AM7020, Ambion, Applied Biosystems, CA, USA) for gene and protein expression quantification and (ii) buffered formaldehyde for immunohistochemistry (IHC) studies. In addition, soon after euthanasia, uteri, oviducts and ovaries with the vascular pedicle were obtained in order to be used for the vascular-repletion technique. They were transported to the laboratory in ice-cold phosphate buffered saline (pH 7.2) with 20 µg/mL gentamicin and 2.5 µg/mL amphotericin added.

For a better explanation of the different experiments carried out in the present study, a graphical representation of the experimental design is depicted in Figure 1.

### 2.2. Experiment 1: Evaluation of Vascular Pattern

#### 2.2.1. Vascular Injection-Corrosion Technique

The modified injection-corrosion technique was performed as previously described [40], on 5 oviducts from mares in the FP, and 5 in the MLP. The vascular bed was washed with distilled water and then injected with an acrylic resin (Evoplast-Tensol nº 70, Part A ref 620651A, Part B ref 620651B, Bostik Limited, Strafford, UK). Polymerization of the resin was accomplished at room temperature and the corrosion of the non-vascular tissue was accomplished with hydrochloric acid (H1758, Merck). Next, the vascular cast was washed with degreasing substances (Extran^®^ AP 13, Merck Millipore, Oeiras, Portugal). The samples were observed and photographed (Leica Wild M3Z Stereo Microscope, Zurich, Switzerland).

#### 2.2.2. Microvascular Density Assessment

To evaluate the microvascular density in the different portions of the equine oviduct throughout the estrous cycle (FP, *n* = 10; ELP, *n* = 10; MLP, *n* = 10), 4-µm paraffin block histological sections from formaldehyde-fixed tissue were stained with anti-CD31 monoclonal antibody (M0823, Clone JC70A, DAKO-Agilent, Santa Clara, CA, USA) in the dilution of 1:50. Negative controls were performed by replacing the primary antibody with mouse IgG (used at the same concentration as the primary antibody, 550878, BD Bioscience, ENZIfarma, Oeiras, Portugal), or by 0.1M phosphate-buffered saline (PBS; pH 7.4).

The anti-CD31 antibody stains the continuous endothelial cells brown, including the capillaries, thus allowing for a better visualization of the vascular areas. For each oviduct portion (ampulla, infundibulum and isthmus), the number of vessels and microvascular areas were determined on histological sections, on 10 randomly chosen microscope fields, using a light microscope at a total magnification of 400×, connected to a computerized cell analysis system (CAS, Becton and Dickinson, Erembodegem, Belgium). All blood vessels were evaluated equally without distinguishing their nature (arterioles, venules and capillaries). Vascular area was calculated as the percentage of the area occupied by blood vessels with respect to the total area in each microscopic field on 10 fields evaluated per each mare. Vessel numbers were also assessed on the same histological sections used for microvascular area determination. Total vascular area and blood vessel number were determined as the mean values for all 10 microscopic fields of each portion of the oviductal tissue evaluated for each mare [37,38].

### 2.3. Experiment 2

#### 2.3.1. Gene Transcription Analysis

Evaluation of the gene transcription of *FGF1*, *FGF2*, *FGFR1*, *FGFR2*, *VEGF* and *KDR* (Table 1) was performed using real-time PCR (qPCR) in the infundibulum, ampulla and isthmus from oviducts obtained at different phases of the estrous cycle (FP, *n* = 5; ELP, *n* = 5; MLP, *n* = 5). From the different portions of the mare oviduct, mRNA was extracted using an mRNA Extraction and Purification Kit (28704; Qiagen, Hilden, Germany), including a DNA-digestion step with an RNase-free DNase Set (50979254; Qiagen), following the manufacturer’s instructions. Quantification and purity assessment of RNA was carried out using the Nanodrop system (ND 200C; Fisher Scientific, Hampton, PA, USA). The observed ratio of absorption at 260/280 nm was approximately 2, and the ratio 260/230 nm was between 2.0 and 2.2 for all analyzed samples. In addition, visualization of 28S and 18S rRNA bands after electrophoresis through a 1.5% agarose gel and red staining (41003; Biotium, Hayward, CA, USA) was accomplished. Reverse transcription was performed with the Reverse Transcriptase Superscript III enzyme (18080093; Invitrogen, GIBCO BRL, Carlsbad, CA, USA) from 400 μg total RNA in a 20 μL reaction volume using oligo(dT) primer (27-7858-01; GE Healthcare, Buckinghamshire, UK), as previously described [41,42]. Specific primers for *FGF1*, *FGF2*, *FGFR1*, *FGFR2*, *VEGF* and *KDR*, as well as reference genes (Table 1) were designed using Primer3 Software and confirmed with Primer Express^®^ (Applied Biosystems, Foster City, CA, USA). To choose the most stable internal control gene under our experimental conditions, five potential genes—β2-microglobulin (*β2M)*, glyceraldehyde 3-phosphate dehydrogenase (*GAPDH*), succinate dehydrogenase complex flavoprotein subunit A (*SDHA*), mitochondrial ribosomal protein L32 (*MRPL32)* and β-actin—were tested. During the validation process, samples from the oviduct (infundibulum, ampulla and isthmus) and endometrium (*n =* 4) from distinct stages of the estrous cycle (FP, ELP and MLP), were run in parallel for the tested genes. The mRNA transcription of *β2M* for oviduct tissues was the most stable reference gene and was unaffected by the experimental conditions [43]. After optimization, the selected primer concentration was 80 nM for target and reference genes, which were run simultaneously. Real-time PCR assays were performed in duplicate wells on a StepOnePlus™ Real-Time PCR System (Applied Biosystems, Warrington, UK), using Power SYBR Green Master Mix (4367659; Applied Biosystems) and the universal temperature cycles (95 °C denaturation for 10 min, followed by 40 cycles of 95 °C for 15 s and 60 °C for 1 min; dissociation step: 95 °C for 15 s, 60 °C for 30 s and 95 °C for 15 s). The specificity of the PCR products was confirmed on 2.5% agarose gel (BIO-41025; Bioline, Luckenwalde, Germany) and by melting curve analysis. The real-time PCR miner algorithm was used for relative mRNA quantitative analysis [44]. For each sample, average cyclic threshold (Cq) levels were related with primer efficiency level (E) using the equation (1/(1þE)Cq) [44]. Target genes’ transcription levels were normalized against that of the reference gene.

#### 2.3.2. Relative Protein Abundance

Western blot analyses for relative protein abundance were performed as previously described [41] on the infundibulum, ampulla and isthmus from 15 mares in different phases of the estrous cycle (FP, *n* = 5; ELP, *n* = 5; MLP, *n* = 5). Briefly, oviduct samples were minced and placed in ice-cold RIPA buffer (50 mM Tris-HCl, pH 7.4, 50 mM EDTA, 150 mM NaCl and 1% Triton X-100) with a protease inhibitor cocktail (Complete Mini Protease Inhibitor Cocktail Tablets, 1 tablet per 10 mL of buffer; Roche) and homogenized on ice. Afterwards, protein concentration was assessed using the Bradford reagent (500-0006; Bio-Rad, Hercules, CA, USA). The amount of total protein used was 40 µg, separated by SDS-PAGE (8% acrylamide gel; ref. 161-0155; Bio-Rad, Hercules, CA, USA). Proteins were transferred to nitrocellulose membranes (ref: 1620116; Bio-Rad) as described in [24]. The relative protein abundance of FGFR1, FGFR2 and KDR was measured using specific primary antibodies against FGFR1 (Orb 156864, Biorbyt, Cambridge, UK, dilution 1:500), FGFR2 (SC 6930, Santa Cruz Biotechnology, Dallas, USA, dilution 1:500) and KDR (Orb 99143, Biorbyt, Cambridge, UK, dilution 1:250). To normalize the loaded protein, a mouse monoclonal antibody against β-actin (A5441, Sigma-Aldrich, Lisbon, Portugal) was used at the dilution 1:10,000. All the membranes were incubated with the primary antibody overnight at 4 °C, except against β-actin, which was incubated for 1.5 h at room temperature (RT). Membranes first incubated against FGFR2 were further incubated for 1.5 h at RT with an anti-mouse IgG (Fc specific)-peroxidase antibody produced in goats (A2554, Sigma-Aldrich, Lisbon, Portugal) at 1:10,000. Membranes incubated against FGFR1 and KDR were further incubated with a conjugated anti-rabbit antibody (P0448, Dakocytomation, Carpinteria, CA, USA), at 1:10,000 for 1.5 h at RT. Membranes incubated against β-actin were later incubated for 1 h at RT with horseradish peroxidase (HRP)-conjugated goat anti-mouse (A2554, Sigma-Aldrich, Lisbon, Portugal) at 1:5000. Protein expression was visualized using luminol-enhanced chemiluminescence (Super Signal West Pico, 34077; Thermo Scientific, Waltham, MA, USA) and image acquisition was performed using a ChemiDoc XRS+ system (Bio-Rad Laboratories, Inc., Amadora, Portugal). Sample. 

### 2.4. Statistical Analysis

GraphPAD PRISM Software was used to analyze data (Version 5.00, GraphPad, San Diego, CA, USA). After having verified that there was no normal distribution of the data with the D’Agostino and Pearson test, a Kruskal–Wallis statistical analysis was performed. To pinpoint specific significant differences between the studied parameters, Dunn’s non-parametric multiple-comparison test was used. Significance was defined as *p* < 0.05. Results are shown as mean ± SEM.

## 3. Results

### 3.1. Experiment 1: Evaluation of Vascularization Pattern

#### 3.1.1. Vascular Injection-Corrosion Technique

To assess the vascular distribution and the type of branching of the uterine artery along the oviduct, corrosion casts were observed. The uterine artery presents a sinuous trajectory along the broad ligament of the uterus. When reaching the oviduct, the oviductal artery progresses from the isthmus towards the infundibulum and its fimbria, showing a quite coiled course. The ampulla depicts sequential dichotomic divisions of vessels that give rise to small arterioles, which are numerous in the infundibulum and the fimbria and which exhibit a brush ramification pattern (Figure 2).

#### 3.1.2. Microvascular Density 

Microvascular structures of the oviduct were assessed on histological sections immunostained brown with an antibody against CD31 (Figure 3).

The vascular areas and vessel counts were evaluated. When all the different regions of the mare oviduct (infundibulum, ampulla and isthmus) were compared, it was noted that in the follicular phase, the isthmus presented the largest vascular area and the highest number of vascular structures with respect to the other areas of the oviduct (*p* < 0.05; Figure 4A,B, respectively). However, this was not observed either in the early or in the mid-luteal phases (Figure 4C–F).

### 3.2. Experiment 2

#### 3.2.1. Transcription of Angiogenic Growth Factors and Their Receptors 

Some angiogenic growth factor (*FGF1*, *FGF2* and *VEGF*) and receptor (*FGFR1*, *FGFR2* and *KDR*) transcripts were studied in the infundibulum, ampulla and isthmus of mares in distinct phases of the estrous cycle. No difference was found in the transcription levels of *FGF1* throughout the estrous cycle (Figure 5A). In contrast, when data were analyzed without distinction between the estrous cycle phases, but considering the different segments of the oviduct, an upregulation of *FGF1* was observed both in the ampulla and the isthmus (*p* < 0.05; Figure 5B). Moreover, the transcription pattern of *FGF2* was also the highest in the isthmus (*p* < 0.05; Figure 5D), but no difference was observed throughout the estrous cycle (Figure 5C). Regarding *VEGF*, no difference was observed either in its transcription throughout the estrous cycle, or between the portions of the oviduct (*p* > 0.05; Figure 5E,F).

As observed for the ligand (Figure 5A), the estrous cycle did not influence the transcription of *FGFR1* (*p* > 0.05; Figure 6A), whereas there was a rise from the infundibulum to the isthmus (*p* < 0.05; Figure 6B). With respect to *FGFR2*, no difference in its transcription was detected either during the estrous cycle or among oviduct portions (*p* > 0.05; Figure 6C,D). When the estrous cycle influence was disregarded, *VEGFR-2* (*KDR*) presented the highest transcription in the isthmus when compared with the infundibulum or the ampulla (*p* < 0.05; Figure 6F), but when the estrous cycle was considered, regardless of the oviduct portions, no difference on transcription was detected during the estrous cycle (*p* > 0.05; Figure 6E).

#### 3.2.2. Relative Protein Abundance of Angiogenic Growth Factor Receptors

The relative protein abundance of FGFR1, FGFR2 and VEGFR2 (KDR) was assessed by means of Western blot (WB) analysis in different portions of the mares’ oviducts. Both FGFR1 and KDR presented the highest protein expression in the isthmus compared to that in the infundibulum or in the ampulla (*p* < 0.05, Figure 7A,B,E,F), whereas no changes between the oviduct portions were found for the FGFR2 gene (*p* > 0.05, Figure 7C,D).

## 4. Discussion

In the internal genitalia of mares, the reproductive organs, such as the ovary and endometrium, present different microvascular changes throughout the estrous cycle. Specifically, the endometrium has not shown any differences in microvascular density [16,37]. In contrast, the corpus luteum has shown differences in its microvascular density, according to progesterone production [38]. To the best of our knowledge, the present work shows for the first time that the area and number of microvascular structures may show changes in specific regions of the mare oviduct (infundibulum, ampulla, isthmus), according to estrous cycle phases. Particularly in the follicular phase, under the influence of estrogen, the vascular bed increases in the isthmus, when compared to the other two regions of the oviduct. In fact, in the rabbit, in the follicular phase, when the oocyte reaches the ampullary isthmic junction, a dilatation of the isthmic sub-serosal venous plexus occurs [45]. In this species, during pregnancy, an increase in the microvasculature was observed, probably due to increased levels of circulating placental hormones [46]. Thus, in an attempt to understand the observed changes in microvascular density in equine oviduct, and due to the importance of angiogenesis in reproductive organs, the expression of the most important angiogenic factors and receptors were evaluated in the mare oviduct.

As appropriate vascularization, adapted to the distinct functional states of the female reproductive tract, is needed to assure adequate reproductive function, several processes involving angiogenesis modulate the vascular bed present in the reproductive organs. It has been observed that angiogenesis is mainly driven by the specific tissue needs in terms of nutrients and oxygen [47]. Our work has shown an increase in the vascular area and structures of the equine oviduct isthmus, although without taking into consideration the nature or caliber of blood vessels. This increase was noted in the follicular-phase oviduct, with a concomitant higher transcription of *FGF1*, particularly in the ampulla and in the isthmus. Interestingly, in the isthmus, we observed coiled first-order arteries that sprout from the uterine artery, which under the action of pro-angiogenic factors and due to their caliber, may contribute to the needs of the vascular bed of that area, preparing and providing the ideal conditions for gamete fertilization. From this area onwards, in the direction of the ampulla and lastly in the infundibulum, tuffs of arteries of small caliber appear as the result of dichotomic pattern ramifications. The geometry of vascular networks may be related to morphological architecture and local microenvironment [48]. The different tri-dimensional architecture of the vessels in the three portions of the oviduct might be associated with different circulatory needs and functions of oviduct areas. As reported in humans and rabbits, microvasculature among the different portions of the oviduct may be related to specific physiological roles of this organ, such as fertilization and embryo transport, enabling a crosstalk between the various regions of the oviduct, ovary and uterus [36].

In the oviduct, the VEGF system presents differences among species in terms of its transcription and relative protein abundance, between portions, and is estrous-cycle-dependent [10,14,49]. These differences suggest various roles in oviduct function, either related with angiogenesis, secretion of oviductal fluid or oviduct contractibility, as has been suggested for swine, cow and human oviducts [10,14,31,49,50], which are crucial for gamete transport, fertilization and embryo transport [31,51]. In addition to the role of VEGF in angiogenesis, in the cow endometrium, VEGF may also stimulate PGF_2α_ production, which is known to trigger luteolysis and a new estrous cycle [49]. In the present study, transcription of *VEGF* in the equine oviduct did not change between portions, which is agreement with observations in the cow oviduct [49]. In contrast, the human oviduct presented a higher transcription in the infundibulum and ampulla, particularly in the peri-ovulatory period [14]. In the swine oviduct, different results from ours were also observed, as *VEGF* showed a higher transcription in the ampulla, compared to the isthmus [10]. However, in the present study, even though *VEGF* transcripts were present in the three portions of mare oviduct and in all phases of the estrous cycle, their specific role in oviduct angiogenesis, motility, fluid secretion or vascular permeability is yet to be proven.

In general, the VEGF receptor KDR is a protein associated with vasculogenesis, angiogenesis, cell proliferation and vascular permeability [52,53]. In tracheal capillaries of mice, after VEGF signaling was blocked with a VEGF-receptor tyrosine kinase inhibitor, a 30% decrease in capillaries was observed 21 days afterwards [54]. In addition, in human cancer, neutralizing monoclonal antibodies against VEGF and small-molecule tyrosine kinase inhibitors targeting VEGFRs has been shown to block its angiogenic activity, resulting in tumor vascular regression, anti-tumor effects and improvements in patient survival [55]. In the sow oviduct, the transcription of *KDR*, either in the ampulla or in the isthmus, rose in the early- and late-luteal phases [10]. Nevertheless, in the human oviduct, *KDR* transcripts were higher in the ampulla and infundibulum compared to the isthmus and thus may be important in the temporal regulation of oviductal secretion [50]. In the present work, the transcription and relative protein abundance of VEGF receptor (KDR) increased in the isthmus of the mare oviduct when the estrous cycle phase was not considered. These data on the isthmus are consistent with the fact that the highest microvascular density was observed in this region of the oviduct in the follicular phase. However, to the best of our knowledge it has not yet been demonstrated that this receptor has a direct role in angiogenesis in the oviduct. 

Like other angiogenic/growth factors, the FGF family is involved in several processes, including cell growth, proliferation, differentiation, and cell survival [56]. In the bovine ovarian follicle, FGF family members are involved in folliculogenesis, especially during the final stage of the follicular phase through stimulation of angiogenesis and granulosa cell survival and proliferation [35]. As shown in other reproductive endocrine organs, FGF has been localized in the uterus of mated and unmated gilts [57], in conceptus tissues [58], and may stimulate ovine luteal cell proliferation [59]. In gilts, FGF2 has specifically been identified in the endometrial epithelium, stroma and myometrium during the estrous cycle and early pregnancy, but without any differences on its expression [57]. Nevertheless, another research group, using immunolocalization techniques in swine, reported an increase in FGF2 in endometrium luminal epithelium and stroma on days 12 and 14 of gestation [58]. In the swine, bovine, and mouse oviduct, the presence of the FGF system has also been reported [15,60,61]. In the murine, the existence of FGF2 in the uterus, oviduct (isthmus and ampulla), cumulus oophorus-oocyte complex, and its receptors in testicular germ cells and in sperm recovered from the cauda epididymis, also suggests the involvement of the FGF system in the *in vivo* regulation of sperm function [61]. Nevertheless, to the best of our knowledge, no references were found regarding the expression of this angiogenic factor in the mare oviduct. In the present study on the equine oviduct, in contrast to *VEGF*, although *FGF1* presented the highest transcription in both the isthmus and ampulla, *FGF2* mRNA levels were more expressed in the isthmus. Nevertheless, only *FGFR1* showed higher transcription and relative protein abundance in the isthmus when compared to the other oviduct portions. These results suggest different roles of the ligands and their evaluated receptors. As observed for FGFR1, KDR also showed higher transcription and protein expression in the isthmus, which is the portion with the highest microvascular density in the follicular phase, compared to the other oviductal regions. These findings might suggest a role of the ligands FGF and VEGF, mediated by their specific receptors, in oviductal function. In fact, when comparing the protein expression of all oviduct portions, one may speculate that the receptor FGFR1 in the isthmus might modulate the action of these growth factors in terms of blood vessel development, oviductal secretion or motility. In contrast, the protein expression of FGFR2 did not differ between the three portions of the oviduct. As such, these data highlight the likely importance of the FGF system in creating the proper conditions for physiological events in the follicular phase in the oviduct [62].

## 5. Conclusions

In conclusion, this work shows that the equine oviduct presents differences in microvascular density in its three portions. In addition, the angiogenic factors VEGF, FGF1, FGF2 and their respective receptors are expressed in all studied regions of the mare oviduct, in agreement with microvascular patterns.

## Figures and Tables

**Figure 1 animals-11-01099-f001:**
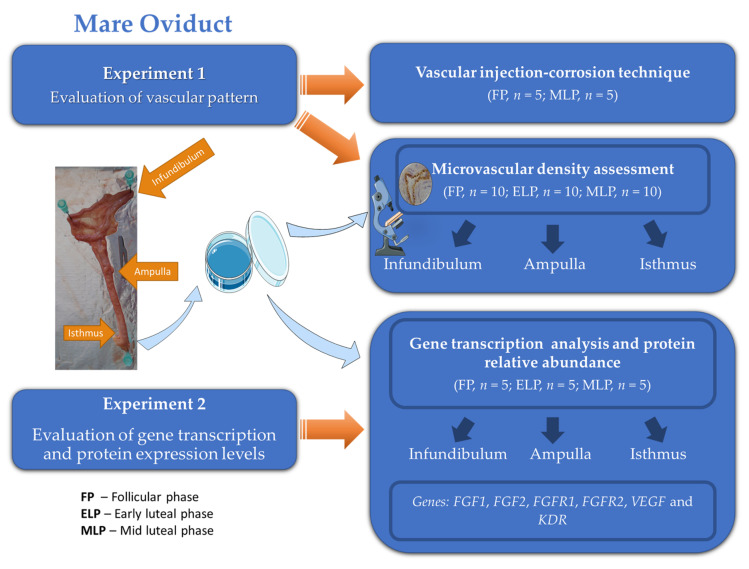
Graphical representation of the experimental design.

**Figure 2 animals-11-01099-f002:**
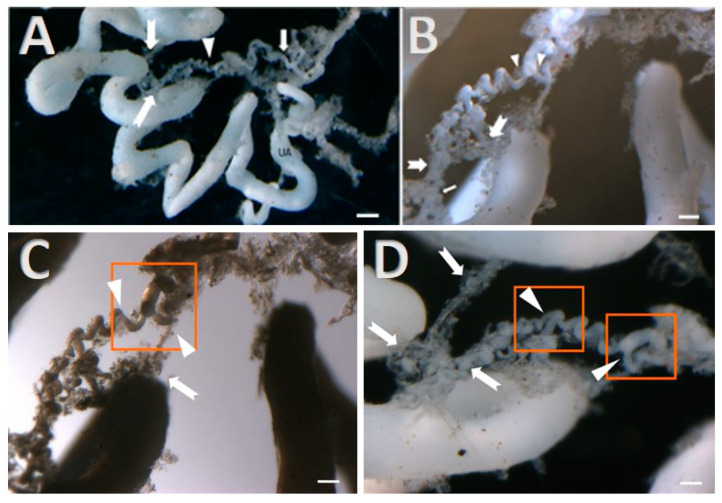
(**A**) Vascularization images of mare oviduct depicted by means of the injection-corrosion technique. On the left panel, one can observe the uterine artery (UA), the isthmus vascularization (arrow), the ampulla vessels (arrowhead) and vascularization of the infundibulum (indented arrow), bar = 0.33 cm. (**B**) Magnification of the arteries in the ampulla shows that a pattern of dichotomic division is present. These vessels go further into small arterioles (indented arrows) in a coiled configuration, providing vascularization of the infundibulum and fimbria, where ramifications are of the brush or comb type, bar = 0.15 cm. (**C**) Visualization by transillumination shows dichotomy divisions of vessels in the ampulla (orange box, head arrows), bar = 0.15 cm. (**D**) Vessels in the infundibulum are small arterioles that split once again in a dichotomous way (orange square) and process as tuffs of vessels that sprout laterally or in a terminal way in the fimbria (indented arrows), bar = 0.15 cm.

**Figure 3 animals-11-01099-f003:**
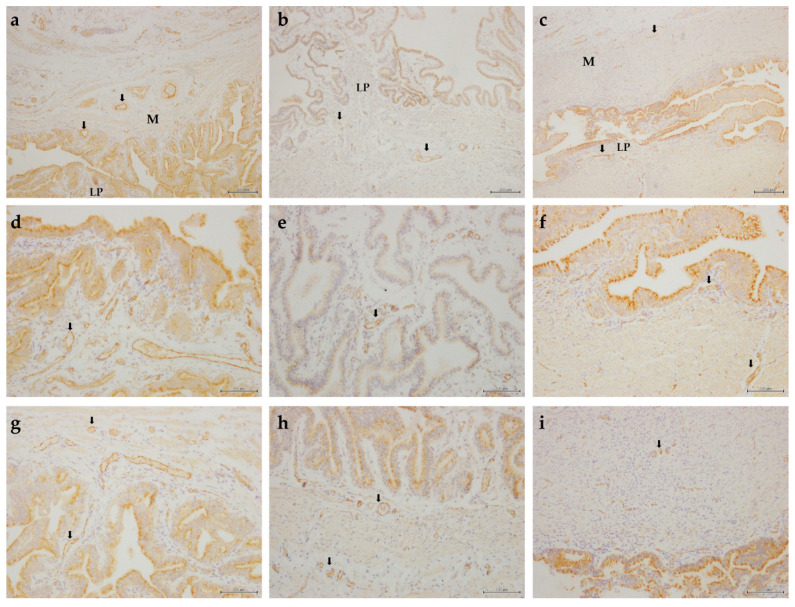
Representative images of vascular structures in the equine oviduct immunostained with anti-CD31 antibody. Infundibulum (**a**,**d**,**g**); ampulla (**b**,**e**,**h**); isthmus (**c**,**f**,**i**). Scale bars of 200 µm (**a**–**c**) and 100 µm (**d**–**i**). Black arrows indicate examples of CD31-positive vessels. M—muscularis; LP—lamina propria.

**Figure 4 animals-11-01099-f004:**
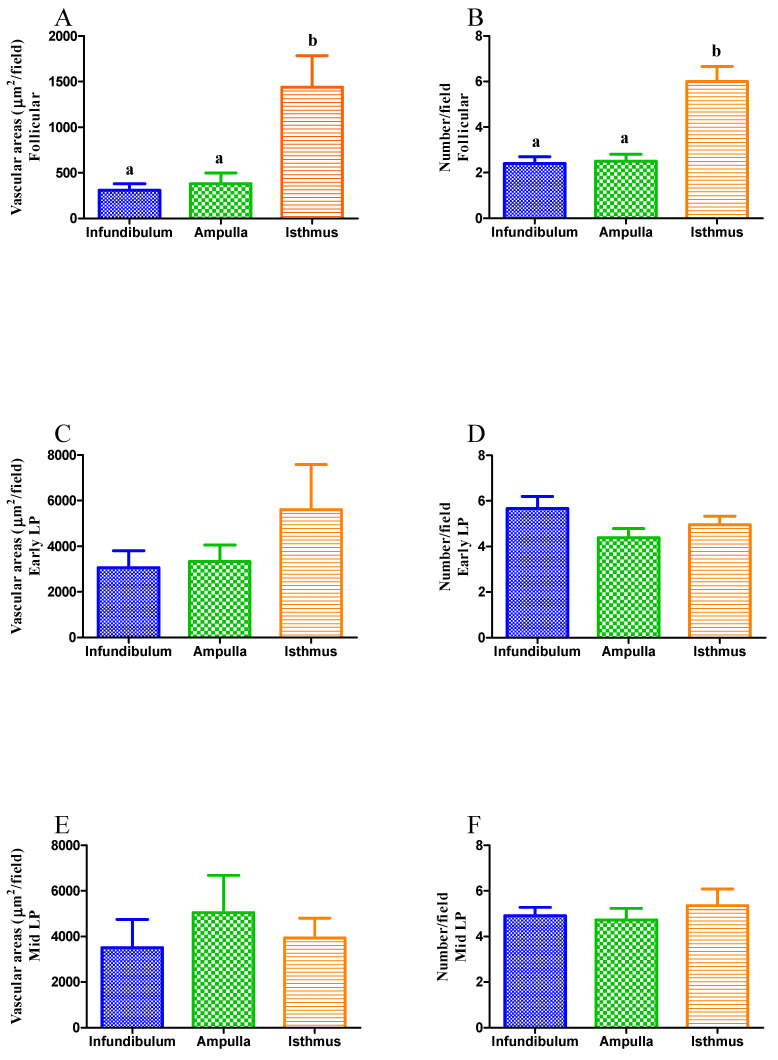
Vascular areas present in the oviduct. Total area of microvascular structures (**A**,**C**,**E**) and total vascular structures (**B**,**D**,**F**) present in 10 random microscopic fields. Analysis was performed considering samples from each estrous cycle phase: follicular (**A**,**B**), early-luteal (**C**,**D**), and mid-luteal phases (**E**,**F**). Bars represent mean ± SEM. Different letters (a,b) indicate significant differences (*p* < 0.05).

**Figure 5 animals-11-01099-f005:**
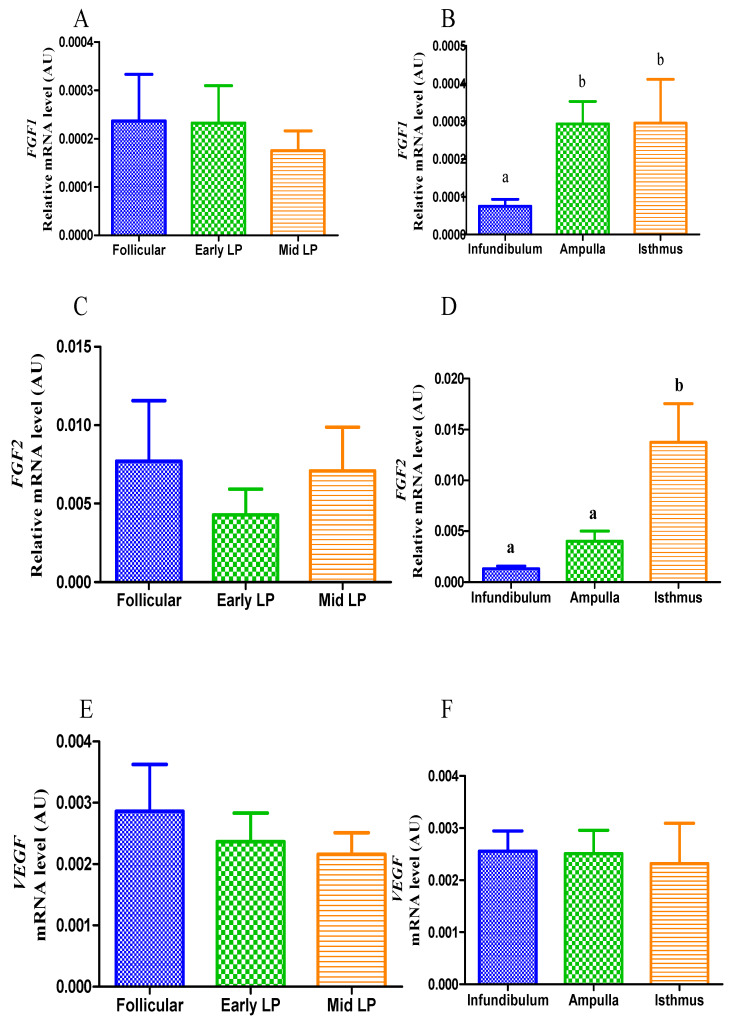
Relative quantification of *FGF1*, *FGF2* and *VEGF* transcripts in the equine oviduct (*n* = 5 samples for each estrous cycle phase; *n* = 5 for each portion of oviduct analyzed). Comparison of transcripts between the follicular phase, early-luteal phase (Early LP), and mid-luteal (Mid LP) (**A**,**C**,**E**). Comparison of transcripts between oviduct portions (infundibulum, ampulla, isthmus), regardless of the phase of the estrous cycle (**B**,**D**,**F**). Bars represent mean ± SEM. AU: arbitrary units. Different letters indicate significant differences (*p* < 0.05).

**Figure 6 animals-11-01099-f006:**
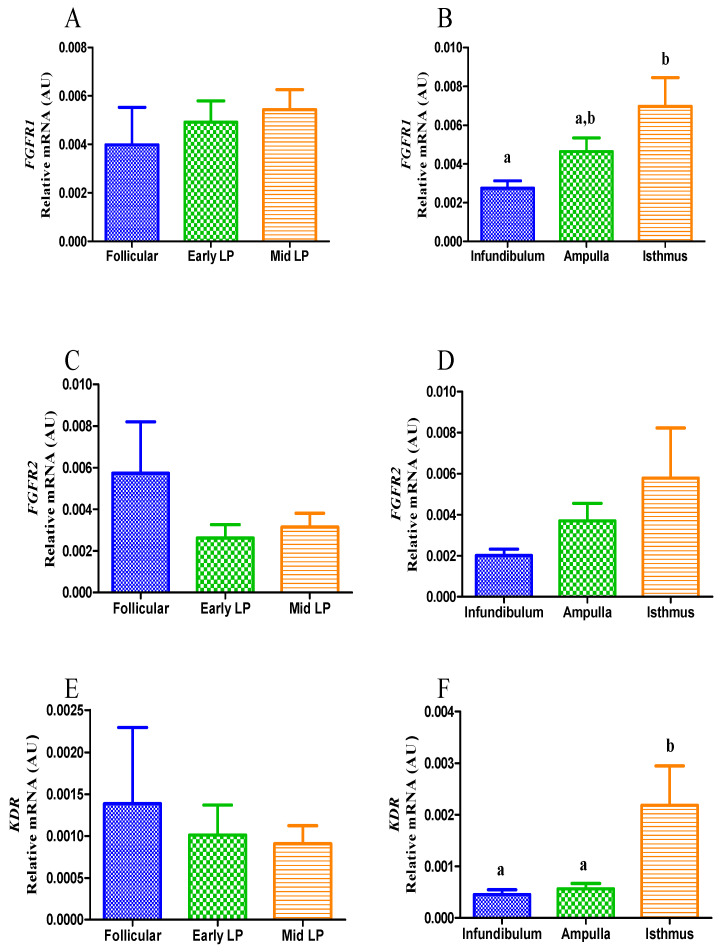
Relative quantification of *FGFR1*, *FGFR2* and *KDR* transcripts in the equine oviduct (*n* = 5 samples for each estrous cycle phase; *n* = 5 for each portion of oviduct analyzed). Comparison of transcripts between the follicular phase, early-luteal phase (Early LP), and mid-luteal (Mid LP) (**A**,**C**,**E**). Comparison of transcripts between oviduct portions (infundibulum, ampulla, isthmus), regardless of the phase of the estrous cycle (**B**,**D**,**F**). Bars represent mean ± SEM. AU: arbitrary units. Different letters indicate significant differences (*p* < 0.05).

**Figure 7 animals-11-01099-f007:**
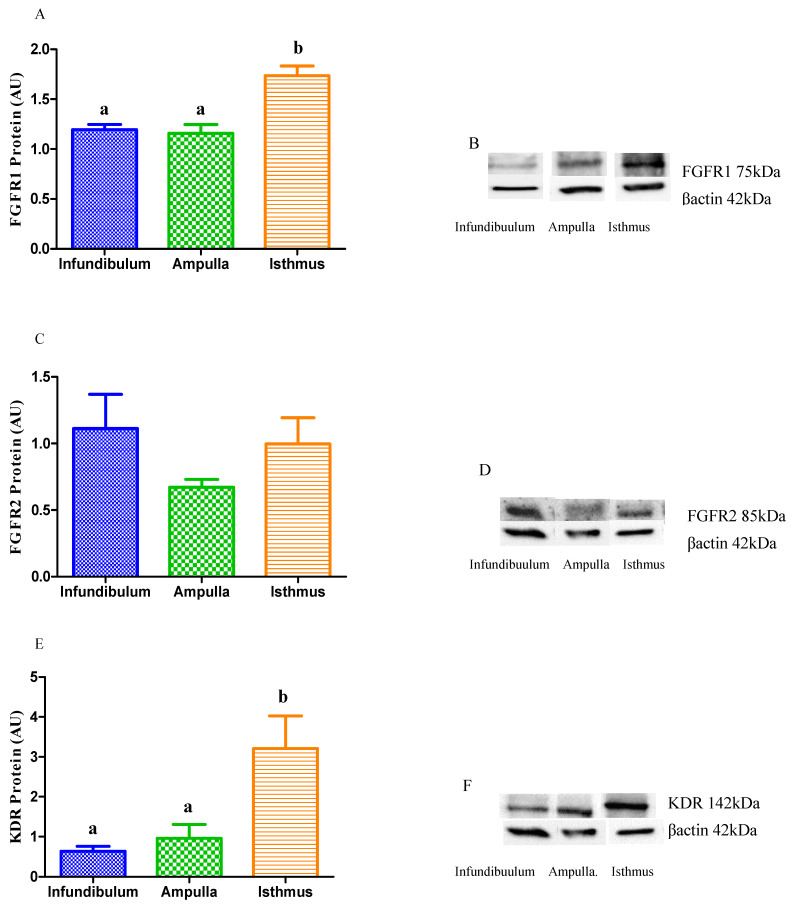
Relative protein abundance of FGFR1, FGFR2 and KDR in the equine oviduct (*n* = 5 samples for each estrous cycle phase; *n* = 5 for each portion of oviduct analyzed). Comparison of relative protein abundance between oviduct portions (infundibulum, ampulla, isthmus), regardless of the phase of the estrous cycle (**A**,**C**,**D**), and their respective representative Western blot bands (**B**,**D**,**F**). Bars represent mean ± SEM. AU: arbitrary units. Different letters indicate significant differences (*p* < 0.05).

**Table 1 animals-11-01099-t001:** Primer sequences used for gene transcription analysis by real-time PCR.

Gene (Accession Number)	Sequence 5′–3′	Tm (°C) ^a^	Amplicon (Base Pairs)
*FGF1* (XM_005599133)	Forward: GTGGATGGGACAAGGGACAG	58.8	187
	Reverse: GGTTTTCCTCCAGCCTTTCC	58.6	
*FGF2* (NM_001195221)	Forward: GGAGAAGAGCGACCCTCACA	59	234
	Reverse: ATACTGCCCCGTTCGTTTCA	59	
*FGFR1* (XM_014736560)	Forward: ACCCAACCGTGTGACCAAAG	59.3	260
	Reverse: GGTTGTGGCTGGGGTTGTAA	59.3	
*FGFR2* (XM_014732956)	Forward: CCAGCTCCTCCATGAACTCC	58.3	237
	Reverse: TGACTGCTTCCTTGGGCTTC	59	
*VEGF* (NM_001081821)	Forward: ATGCGGATCAAACCTCACCA	59.9	117
	Reverse: AGGCCCACAGGGATTTTCTT	58.5	
*KDR* (XM_014738773)	Forward: CTTCCAGTGGGCTGATGACC	59.1	100
	Reverse: AGCTTCCACCGAAGATTCCA	58.3	
*β2M*	Forward: CGGGCTACTCTCCCTGACTG	58.5	92
(X69083)	Reverse: TTGGCTTTCCATTCTCTGCTG	59	

*FGF1*: fibroblast growth factor 1; *FGF2*: fibroblast growth factor 2; *FGFR1*: fibroblast growth factor receptor 1; *FGFR2*: fibroblast growth factor receptor 2; *VEGF*: vascular endothelial growth factor; *KDR*: kinase insert domain receptor (vascular endothelial growth factor receptor 2); *β2M*: beta 2 microglobulin. ^a^ Tm: temperature of the primer calculated using the nearest neighbor algorithm.

## Data Availability

Data will be available upon request to the corresponding author.

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
