# Peer review of "Microvascularization and Expression of Fibroblast Growth Factor and Vascular Endothelial Growth Factor and Their Receptors in the Mare Oviduct"

_animals, 2021, doi:10.3390/ani11041099_

Round 1
Reviewer 1 Report
Dear authors,
This work analyses the gene and protein expression, and their receptors of different factors related to vascularization in mare oviduct. The manuscript is well structured and written, the introduction provides sufficient background to understand the relevance of the study, the research design is appropriate, the methods are adequately realized and described, the results are clearly presented, and the conclusions are supported by the results. In summary, this study is a good research work, well presented and with relevant results. Only minor changes related with the format are necessary:
- The authors should review the formatting of transcription factors (should always be italicized) and proteins (not italic), for example, lines 92-98, line 391, or line 415.
- Line 232: eliminate italic letter in “1:5,000”.
- Line 404: add point after reference 54.
I would like to congratulate you on the research work carried out.
Author Response
Reviewer #1
We are very grateful to you for the valuable, thorough, and very constructive comments, and for the time spent evaluating this manuscript. We do hope we have addressed all of your concerns appropriately.
- The authors should review the formatting of transcription factors (should always be italicized) and proteins (not italic), for example, lines 92-98, line 391, or line 415.
- Thanks. This was corrected.
- Line 232: eliminate italic letter in “1:5,000”.
- Thanks. This was corrected.
- Line 404: add point after reference 54.
- Thanks. This was corrected.
Reviewer 2 Report
The revised version of the manuscript has addressed most of the comments and suggestions of the previous review.
Author Response
Reviewer #2
We are very grateful to you for the valuable, thorough, and very constructive comments, and for the time spent evaluating this manuscript again. We do apologize for giving you all the extra of reevaluating this article.
This manuscript is a resubmission of an earlier submission. The following is a list of the peer review reports and author responses from that submission.
Round 1
Reviewer 1 Report
Dear authors,
This work analyses the expression and presence of proteins related to angiogenesis in different portions of the mare’s oviduct. The manuscript is well written and structured, but some major revisions are necessary:
- In the keywords, the authors should eliminate numbers and abbreviations. They should add the complete name of factors and their receptors.
- In all the manuscript, the authors should separate correctly the references with comma. For example, lines 69, 70, 72, and others.
- Line 74: If other exist, the authors should list them and add references.
- Line 91: The authors refer to activate the expression genes? What mean “activate”? If these are genes, please, italized.
- Line 96: This is a member of FGF family? Please, explain it.
- The authors should eliminate some spaces in all manuscript (for example, lines 128, 147, and others).
- The authors should unify the name of beta-actine (lines 177, 179, 184…)
- Lines 196-197: If the authors evaluated the integrity of the RNA using Nanodrop, they should indicate that, and they should indicate the values obtained. Only the visualization of 28S and 18S rRNA bands in electrophoresis does not ensure the quality and integrity of the RNA.
- Lines 210-212: The most stable gene in for oviduct was beta2M, but, in the others tissues analyzed? explain it.
- Lines 213: change “SYBER” to “SYBR”.
- Line 219: change “transcription” to “transcription”.
- Statistical analysis: before to use a parametric test as ANOVA, the authors should verify the normality and homoscedasticity of the data. Normally, gene expression data do not usually follow a normal distribution, so it is necessary to apply a test such as ANOVA, perform a previous mathematical transformation, usually logarithmic.
- In discussion, the manuscript needs more references that support some sentences (for example, lines 331-333, or lines 361-362).
- Lines 375-376: the authors should review the bibliography. Expression of FGF2 has been found in bovine, murine, and porcine oviduct.
I hope these recommendations improve the quality of manuscript.
Author Response
Reviewer #1
We are very grateful to you for the valuable, thorough, and very constructive comments, and for the time spent evaluating this manuscript. All the corrections suggested were addressed, and are written in red font, throughout the manuscript. We do hope we have addressed all of your concerns appropriately.
- Keywords were corrected.
- References are now separated with a comma.
- L 74: Other references were added.
- L91- The sentence was corrected. Enzymes were activated, not the genes. Gene names were italized.
- L96 – We do apologize – that was a typo. We meant FGF2.
- L128, 147, etc. All extra spaces were deleted.
- Beta-actin was corrected and depicted as β-actin.
- L196-197 – This was clarified.
- L210-212 - β2-microglobulin was the most stable gene for the comparison of gene transcription among the three different portions of the oviduct. This refence gene was also the most stable one for equine corpus luteum transcription studies performed by our group (Galvão et al., 2014; http://dx.doi.org/10.1155/2014/682193). Nevertheless, in mare endometrium we have found that RPL32 should be the refence gene, instead (Rebordão et al., 2019 https://doi.org/10.1016/j.domaniend.2018.10.004)
- L213 – This was corrected.
- L219 – This was corrected.
- You are correct. Statistical analysis was performed to verify data normality, as explained in the statistical analysis section. Since no regression analysis was performed, we did not verify data homoscedasticity.
- L331-333 – The specific sentence on lines 331-333 reflects the authors opinion/interpretation of the results. We are sorry but we could not find any references on that. But, we have added some text at the end of the paragraph, and reference [36] that might improve the paragraph.
- L361-362 – Available references were included (now line 383).
- L375-376 -References were added and sentence corrected.
Reviewer 2 Report
This paper describes an original study on the microvascularization and expression of angiogenic factors in the mare oviduct. The topic is interesting because in contrast to other domestic animals very few information is available in the equine species. The overall objective of the work is well presented and justified. A high number of research tools have been used. Anatomical, histological and molecular techniques have been used. The article is well conceived but there are important drawback which need to be amended. The design is a bit confuse, the methodology requires some clarification, the results are not clearly displayed and explained, and some topics in the discussion are too speculative. The conclusion also include information that is not clearly displayed in the results. The paper requires a major review, mainly in some critical issues and minor comments which are being summarized.
Experimental design. To avoid confusion a diagram is required. Also, it is important to clarify how many oviducts were used for each procedure (i.e. vascular injection). Why do the authors talk about two experiments and no mention again in the results or discussion?
Material and methods. 1. Injection of vessels. Did the authors only injected through the uterine artery? The ovarian artery is not mentioned and some vessels supplying the oviduct (infundibulum and ampulla mainly) are directly connected with the ovarian artery. This could have influenced the results, as injecting only from the uterine artery may contribute to preferentially fulfill those vessels supplying the oviduct isthmus. As there are previous works on this topic in other species, it must be clarified and discussed. 2. Assessment of microvascularization was done on 10 randomly chosen fields. It is assumed that fields randomly corresponded to either epithelium, lamina propria or muscular, which introduces quite a long variability between isthmus, ampulla and infundibulum. For instance, the width of propria and muscularis is quite different from those oviduct regions. Again, this source of variability has not been considered and should be reviewed and properly discussed. 3. Statistical analysis is too concise. Did authors check that Anova was te appropriate test to use for all the comparisons? In some cases, a priori, it is difficult to believe that Anova postulates were accomplished with such a low number of specimens (i.e 5 in some cases). What operations were done with GraphPAD? Please explain the statistical section in detail.
Results. 1. Figure 1 it is not possible to see what the authors are explaining in the text. More illustrative pictures are required (different magnifications, different oviduct areas, different specimens). Marking magnification by x10 or 66X is not recommended, please use bars in all pictures (also in histology -Figure 2- where they are absent). 2. Microvascularity. Same problem. More illustrative pictures required (different magnifications, histological regions -propia versus muscular- in each portion. 3. Figure 3 lacks a comparison between stages. It is said that vascularization was higher in the follicular phase but not statistical comparison is done between estrous phases. This is a key topic, because if this is not well demonstrated part of the discussion and conclusion is unsupported by the results. Legend of Figure 3, 4, 5 and 6 does not mention what a, b, c, d.... mean. 4. In the plots, standard deviation instead of SEM is recommended.
Discussion. 1. Lines 315-317. ...."where the vascular bed increases in the follicular phase....." this is not supported by results. Similar problem appears in lines 328-329. 2. First paragraph. Comparing microvascularization results in mare with rabbit only is not enough and does not really clarifies the obtained results. There are studies in the literature from other species more related to this work. Please check and discuss. 3. Second paragraph, lines 329 -339, is too speculative, especially the last sentence. Are authors suggesting that vasculogenesis is a valid angiogenic mechanism in the oviduct?. 4. Discussion must be more rigorous and focused on the real results obtained with statistical significance. 5, there is a lot of discussion about the role of KDR in angiogenesis, but actually it has not been demonstrated yet that this factor has a direct role on angiogenesis in the oviduct.
Conclusion. "and throughout the estrous cycle" is not demonstrated.
Author Response
Reviewer #2
We are very grateful to you, for the valuable, thorough, and very constructive comments, and for the time spent evaluating and managing this manuscript. We hope we have addressed your suggested corrections accordingly. Changes are written in blue font, throughout the manuscript.
Experimental design – Identification of the two experiments were now indicated in the results section, according to Materials and Methods section. In the Experiment II, the sequence of assays was changed in agreement with the results.
To clarify the Experimental Design a diagram was prepared (Fig. 1) , and the number of oviducts are now indicated. By mistake we had not counted the oviducts used for vascular corrosion technique, which were 10 in total, as indicated. Thus, the total number of oviducts used were 40, from 40 mares.
Materials and Methods
Fig. 2 (old Fig. 1) was redone now and shows different areas of the oviduct. We wish we had better photos, but the immune studies were performed in 2010, and no longer have the quality we wished for. We do apologize for that. Bars, as requested were added to Figures.
We are in accordance with the reviewer, as the ovarian artery contributes for the vascularization of the oviduct. Nevertheless, the uterine artery is the main arterial supply of the uterus in all species (Nomina Anatomica Veterinaria 6th edition, 2017). In the particular case of the mare, it arises from the external iliac artery, while the ovarian artery arises in the vicinity of the caudal mesenteric artery, from the abdominal Aorta. For that reason, to a great extent this vessel stays in the animal carcass, when genitalia are retrieved post mortem. We are aware that the calibre of the uterine artery is much higher than the calibre of the ovarian artery. This latter, although a longer vessel, since it arises from the terminal area of the abdominal aorta, before its division in the external iliac vessels, also presents a much smaller calibre and therefore less flow potential. Although we did not have cannulated the ovarian artery, we do believe that with great probability the network of vessels that arise from that artery was also filled with the replective injection material. We injected the acrylic resin very slowly and we stopped the injection only when the material started to appear in the broad ligament of the uterus. So, we do believe that these vessels might have been fulfilled by retrograde way, as well. The reason why we did not inject the replective resins in the ovarian vessel has to deal with the fact the pieces over we performed our work were cut by the slaughterhouse workers, and in most cases it was not possible to properly identify the ovarian vessel.
The Statistical analysis was now performed to verify data normality, as explained in the statistical analysis section.
You are correct when you say that the width of the muscularis, and the lamina propria is different among the regions of the oviduct. In fact, the operator who photographed these different microscopic fields of the mare oviduct was always the same, which contributed for a decrease in the error. Besides, he made sure the photographed area was representative, and occupied most of the microscopic field. Since 10 photos were taken from each portion of the oviduct, from each mare, and in 10 mares, we do believe they correspond to the actual vascular pattern of the different portions in the oviduct, in spite of their different histological characteristics.
Results. Figures were redone and bars were added.
Legends of Figures 4,5,6 and 7 - The meaning of a, b, c, d, are now explained. Thanks.
Discussion: L315-317 -You are correct – thanks for pointing it out to us. This issue was now clarified in the results section. This comparison was done among the three different portions of the oviduct, for each phase of the estrous cycle, independently. Therefore, we believe now the discussion agrees with the presented results.
L328-329- We added some information concerning human oviduct microvascularization.
Concerning the role of KDR in angiogenesis, we added the sentence you suggested.
Discussion was rewritten, mainly focussing on the data with significance,as you suggested. We do hope we have addressed your major concerns.
L329-339 – in order to be more precise, the last sentence was deleted. The text was rearranged, as the different tri-dimensional architecture of the vessels in the three portions of the oviduct might be associated with different circulatory needs and function of oviduct areas.
Conclusions: As requested the sentence “throughout the estrous cycle was deleted.
Reviewer 3 Report
This study investigates the vascular changes of mare oviduct during the estrous cycle. The study is innovative and well conducted. Only small modifications have been indicated.
The reviewer requires the following revisions before acceptance:
- Table 1 shows data slightly different from those reported in the text of the results. Eg. The primary antibodies reported in lines 174 and 175 "FGFR1 (Orb 156864, Biorbyt, Cambridge, UK, dilution 174 1: 250), FGFR2 (SC 6930, Santa Cruz Biotechnology, Dallas, USA, dilution 1: 250)" report dilution values different than those reported in table 1. We therefore ask the authors to modify the text. It is also suggested that Table 1 could be removed as data are currently written in the text.
- In table 2 please modify in " 5´-3´ "
- Line 213 - 214: please, specify primers concentrations, melting temperatures and PCR cycling parameters.
Author Response
Reviewer #3
We are very grateful to you for the valuable, thorough, and very constructive comments, and for the time spent evaluating this manuscript. All the corrections suggested were addressed, and are written in green font, throughout the manuscript. We do hope we have addressed all your concerns appropriately.
The “old” Table 1 you referred was removed. Information regarding primary antibodies concentrations was now corrected.
Table 2 – (currently Table 1) -This was corrected.
L213-214 – Primers concentrations, and PCR cycling parameters were added to the text. Melting temperatures were included in a new column in Table 1.